# Does Place Matter? An International Comparison of Early Childhood Development Outcomes between the Metropolitan Areas of Melbourne, Australia and Montreal, Canada

**DOI:** 10.3390/ijerph16162915

**Published:** 2019-08-14

**Authors:** Catherine Dea, Lise Gauvin, Michel Fournier, Sharon Goldfeld

**Affiliations:** 1Department of Social & Preventive Medicine, École de santé publique, Université de Montréal, Montréal, QC H3N 1X9, Canada; 2Direction Régionale de Santé Publique, Centre Intégré Universitaire de Santé et Services Sociaux du Centre-Sud-de-l’Île-de-Montréal, Montréal, QC H2L 1M3, Canada; 3Health Innovation and Evaluation Hub, Centre de recherche du CHUM, Montréal, QC H2X 0A9, Canada; 4Centre for Community Child Health, Murdoch Children’s Research Institute, & Faculty of Medicine, Dentistry and Health Sciences, University of Melbourne, Melbourne VIC 3052, Australia

**Keywords:** early childhood development, child health inequities, early development instrument, children’s health and well-being, international comparison, Melbourne (Australia), Montreal (Canada)

## Abstract

There is strong consensus about the importance of early childhood development (ECD) for improving population health and closing the health inequity gap. Environmental features and public policies across sectors and jurisdictions are known to influence ECD. International comparisons provide valuable opportunities to better understand the impact of these ecological determinants on ECD. This study compared ECD outcomes between metropolitan Melbourne (Australia) and Montreal (Canada), and contrasted disparities across demographic and socioeconomic characteristics. *Methods:* Population wide surveys using the Early Development Instrument (EDI) were conducted among 4–6 years-old children in both Montreal and Melbourne in 2012, measuring five domains of ECD: 1-Physical Health/Well-Being (PHYS); 2-Social Competence (SOC); 3-Emotional Maturity (EMOT); 4-Language/Cognitive Development (COGN); and 5-Communication Skills/General Knowledge (COMM). Descriptive analyses of summary EDI indicators and domain indicators (including median scores and interquartile ranges) were compared between metropolitan areas, using their respective 95% confident intervals (CIs). Analyses were performed using Stata software (v14). *Results:* The proportion of children developmentally vulnerable in at least one domain of ECD was 26.8% (95% CIs: 26.2, 27.3) in Montreal vs. 19.2% (95% CIs: 18.8, 19.5) in Melbourne. The Melbourne advantage was greatest for EMOT and COGN (11.5% vs. 6.9%; 13.0% vs. 5.8%). In both Montreal and Melbourne, boys, immigrants, children not speaking the language of the majority at home, and those living in the most deprived areas were at greater risk of being developmentally vulnerable. Relative risks as a function of home language and area-level deprivation subgroups were smaller in Montreal than in Melbourne. *Conclusion:* This study shows that Melbourne’s children globally experience better ECD outcomes than Montreal’s children, but that inequity gaps are greater in Melbourne for language and area-level deprivation subgroups. Further research is warranted to identify the environmental factors, policies, and programs that account for these observed differences.

## 1. Introduction

There is strong international consensus regarding the crucial importance of early childhood development (ECD) for improving population health and reducing inequities throughout the life course [1,2,3]. Investments in ECD services and policies are also recognized as highly valuable and cost-effective for developing human capital and sustainable economies in all societies [4,5,6]. Besides parenting skills and other family characteristics, determinants of ECD take root in other spheres of influence [7]. For instance, the broader socioeconomic context as well as family policies at the national level are major drivers of ECD [1,8]. Research over the past decades has also revealed that local communities in which children grow up can expose them to a wide variety of protective and risk factors directly associated with ECD [9,10,11].

At the municipal level, there is growing interest among policy makers, their partners, and the public in regards to how cities might improve health and quality of life for families and children [12,13]. In many countries, including Australia and Canada, ‘Healthy Cities’ and ‘Child-Friendly Cities’ initiatives have proliferated [14,15]. For instance, the Victorian Child Friendly Cities and Communities Advisory Group works to bring individuals and communities together in order to improve access to supportive environments and services for children in Victoria (Australia) local councils [16]. In Quebec (Canada), the city of Montreal recently released a new Child Policy with the aim to better support physical, cognitive, emotional, and social needs of children [17]. The objectives of these municipal initiatives generally consist of creating child-friendly environments and of supporting families and their young children’s health and development.

### 1.1. The Greater Montreal and the Greater Melbourne Areas

Montreal (MTL) is the biggest city of the province of Quebec and the 2nd largest city in Canada. The city is located on an island, next to the St. Lawrence River. The metropolitan area also comprises North and South shore suburbs where about 50% of the Greater Montreal area population lives [18]. Melbourne (MEL) is the capital and the biggest city of the state of Victoria in Australia. Its main geographical features include the Yarra River dividing the city into North and South, and its location on Port Phillip Bay [19]. At the time of this study, both metropolitan cities shared similar sociodemographic characteristics for their respective populations (Table 1). [20,21]

### 1.2. Governance Structures and Levels of Action for the Early Years

Both Canada and Australia are constitutional monarchies with a tripartite political structure, each level having legally defined mandates. In Montreal, municipalities, and districts constitute the local government which is responsible for cultural and sport activities (e.g., library, festivals), security issues (e.g., police, firefighting), as well as vast range of built environment features, such as city streets, bike paths, public transport, green spaces, playgrounds, and housing [22]. Although Canadian municipalities are becoming more interested in families’ quality of life, they are not responsible for the funding, planning, or delivery of direct services to children which are under Quebec provincial jurisdiction. The Government of Quebec is responsible for financing and implementing an entire range of programs and policies to better support children’s health and well-being via multiple departments such as public health programs, primary care services, early childhood education and primary schools. Moreover, many social policies (e.g., child benefits, paid parental leave, housing subsidies) targeting children and families are under other provincial departments [23,24]. Furthermore, the Government of Canada (i.e., federal level) has its own roles in early childhood education, health and social policies, such as defining national orientations and frameworks, funding research, transferring money to provinces and providing additional child benefits for families [24,25]. Among other matters, the Government of Canada is also responsible for immigration, indigenous and international affairs as well as other social policies and subsidies (e.g., employment insurance, affordable housing). 

In Australia, a similar political system based on three levels of government plans, delivers or funds all different services for children and families [26]. Local councils (i.e., local government areas) represent the most proximal tier of government and traditionally assume most of the decision making about community facilities and town planning, including libraries, parks, roads, public recreation, etc. [26,27]. Over time the roles of councils have expanded to include provision of services previously under state jurisdiction, such as community health services (including Maternal and Child Health), public transport, and to some extent early childhood education [26,27]. Nevertheless, an important part of funding and planning of health, social, and education services remain under the state’s jurisdiction, similarly to the roles played by provinces in Canada. The Australian Government is also involved in improving child health and well-being in many ways, for instance by developing strategic planning and frameworks, and providing additional programs, services, and benefits to families, for instance child care subsidies and accreditation [26].

### 1.3. Importance of International Comparisons of ECD Outcomes

Although ECD is internationally recognised as a priority target, positively impacting the way children grow and develop remains a tremendous challenge throughout the world [28,29,30]. Just as the extensive monitoring globally in place with limited child health indicators (e.g., children mortality rate, immunization rate), cross-national measures of ECD domains with valid indicators are essential for effective action. High-quality international comparisons are required to better understand the environmental determinants of ECD and implement the most impactful interventions [31,32,33,34,35]. Countries, states, cities, and communities that share similar socioeconomic contexts provide the most robust comparisons as it might be hypothesized that ECD outcomes should be [33] similar as should the extent of inequalities. Such geographical comparisons of ECD outcomes could possibly contribute to improvement in family and child policies and services [28,29,36]. The decision to compare metropolitan MTL and MEL was due to their similar population and economic characteristics. It was anticipated that identification of differences could provide a backdrop for further comparisons which in turn could suggest policy possibilities for universal and targeted programs and policies for young children.

In keeping with this notion, the first objective of this study was to compare ECD outcomes among kindergarten-aged children between the metropolitan areas of MTL and MEL, as measured by a series of EDI indicators. The second objective was to contrast disparities across demographic and socioeconomic groups between both cities.

## 2. Materials and Methods 

### 2.1. Overall Design

This study is an international comparison of ECD outcomes at the metropolitan level in 2 states (Victoria and Quebec) in Australia and Canada using two cross-sectional population surveys that took place in 2012. Both surveys were based on the Early Development Instrument [37,38,39], which will be detailed further below. Informed consent was obtained from schools, teachers, and parents for each survey. In Canada as well as in Australia, high security and confidentiality standards were used for the survey data storage and results dissemination. The research protocol received the approval of Human Research Ethics Committees in both settings (CIUSSS Centre-Sud-de-l’île-de-Montréal DIS-1617-03, Melbourne Royal Children’s Hospital HREC-24051P) for the secondary use of their respective survey data. 

### 2.2. Participants

The study population comprised all children who were in their first year of compulsory schooling in 2012 in the metropolitan areas of MTL and MEL. In both settings, children who had less than four EDI valid domain scores (i.e., no more than 30% missing answers per domain), those younger than 4 years-old or with special needs (SN) were excluded from analyses, as recommended by the instrument authors [39].

### 2.3. Measures

#### 2.3.1. Early Development Instrument (EDI)

The EDI is a validated 103-item questionnaire developed at the Offord Centre for Child Studies (McMaster University, Canada) to measure children’s development during their first year of compulsory schooling [39]. Aggregated EDI profiles are increasingly being used at all jurisdictional levels (local, regional, national), as this instrument enables meaningful comparisons both within and across countries [33,40]. The EDI questionnaire is based on teachers’ reports on all children in their class. The questions measure children’s skills and capabilities to meet age-appropriate expectations across five developmental domains: 1-Physical Health and Well-being (PHYS); 2-Social Competence (SOC); 3-Emotional Maturity (EMOT); 4-Language and Cognitive Development (COGN) and 5-Communication Skills and General Knowledge (COMM). For each domain, a ‘vulnerability’ cut-off is set as being at the 10th percentile score of the reference population. It is noteworthy to mention that the EDI was not designed to screen or diagnose children individually, but rather to track ECD outcomes at a population level over time and across geographic areas [39,40,41]. The EDI may be used by communities, policymakers, public health practitioners, and researchers to better inform the planning of child and family policies and services [29,33,42,43]

Over the last decade, the EDI has been increasingly used across the world as a measure of ECD across different jurisdictions [36,37,44,45,46,47]. Up to now, the EDI has been used in over 25 countries [33,40,44,45,48,49] and minor adaptations have been made for country specific usage to accommodate different languages and cultural contexts. Several validity studies comparing adapted versions of the EDI to the original version of the questionnaire and to other standardised tests have been conducted in numerous countries [48,49,50,51,52]. This research shows high internal consistencies of subscale scores in both original and adapted version of questionnaires across the five domains of development as well as strong concurrent validity. These studies support the relevance of using the EDI over more time-consuming standardised tests of ECD. As the EDI provides a well validated population-based measure at relatively low cost, there is increasing support from different organisations such as the World Bank for its use at the national level, but also as a recommended test for international comparisons of ECD [28,33,36,53].

Since 2004, 12 out of 13 Canadian provinces and territories have been collecting data on the EDI at regular intervals [40,47]. In the province of Quebec where Montreal is located, the EDI was introduced in 2012 with the Quebec Survey of Child Development in Kindergarten (QSCDK). Interestingly though, an initial survey had previously been conducted on the Island of Montreal in 2007 [54]. The original EDI questionnaire was translated into French using the translation-backtranslation method for each item [38,51]. For the first two first administrations of the QSCDK (2012 and 2017), the reference population standards were based on data from the 2012 survey [38,55]. In 2009, Australia became the first country to collect data nationally using the Australian Early Development Index (AEDI) [41,52]. The AEDI (now the Australian Early Development Census (AEDC)) has now been conducted in 2009, 2012, 2015 and in 2018 across the entire country, with the 2009 data serving as the reference population [56]. In the AEDC, the number of questions was reduced from 104 to 95, based on item analysis using a Rasch model on over 4000 children and comparing the adapted questionnaire with data from the Longitudinal Study of Australian Children [50,57]. The nine items could be removed without any loss of scale precision. Both the QSCDK (1st ed.) and the AEDC (2nd ed.) were conducted in 2012, the contemporaneous data collection enabled a synchronous comparison of ECD outcomes between MTL and MEL. 

#### 2.3.2. Covariates

The EDI provides summary indicators and domain indicators of ECD, which are then disaggregated by territory or subgroup. Proportions of children vulnerable in at least 1 domain (% DV1) and those vulnerable in at least 2 domains (% DV2) of the EDI were used in this study as summary indicators. Comparisons were also made specifically for each of the 5 EDI domains with proportions of vulnerable children, median scores and interquartile ranges by domain. Results were stratified by sex, home language, place of birth and area-level deprivation.

#### 2.3.3. Data Harmonization

All outcomes of interest and covariates to be compared between MTL and MEL were directly computed with data sets from their respective survey. Table 2 shows the main methodological differences between both surveys and how those aspects were addressed in this study. Common vulnerability cut-offs were necessary to define which children were vulnerable and thus make a valid comparison of ECD outcomes. MTL’s data were thus re-analysed using the AEDC 2009 10th percentile cut-offs. Age-based cut-offs were used to take into account the age differences between children in MTL and in MEL. The language was categorised as French, English or Other for MTL’s children, whereas as English or Other for MEL’s. In both cities, quintiles of area-level deprivation were computed for family’s disadvantage status. Quintiles were calculated using each setting’s socio-economic index, and were based on the populations of the state of Victoria and the province of Quebec for MEL and MTL respectively. The Material and Social Deprivation Index (MSDI) was used in MTL [58], whereas the Index of Relative Socio-Economic Disadvantage (IRSD) was used in MEL [59]. 

### 2.4. Statistical Analyses 

Descriptive analyses of summary EDI indicators (DV1 and DV2) and domain indicators (including median scores and interquartile ranges) were compared between the metropolitan areas of MTL and MEL, using their respective 95% confident intervals (CIs). Statistically significant differences were defined by no overlap in CIs. To analyse patterns of associations between subgroups and EDI indicators, relative risks (RR), and risk differences (RD) for developmental vulnerability were compared between cities, also using the absence of overlap of CIs. We undertook a number of sensitivity analyses (different vulnerability cut-offs, language groups, and exclusion criteria) to ascertain the robustness of the main findings. Stata v.14 was used for all statistical analyses [62].

## 3. Results

### 3.1. Characteristics of Participants

Characteristics of the children surveyed appear in Table 3 for both metropolitan areas. Overall, children were slightly older in MTL (mean age: 72.1 months) than in MEL (mean age: 69.1 months) at the time of the surveys. In both settings, proportions of immigrant were relatively similar (≈9%) as were the proportions of children speaking at home a language different than that of the majority (≈24%). Regarding area-level deprivation, there were proportionally more children in MTL living in the least deprived quintile of the larger jurisdiction compared to MEL (27.0% vs. 23.3%). Children from the most deprived quintile of the larger jurisdiction were in greater proportion in MTL (19.0%) compared to MEL (16.9%). 

### 3.2. Overall Results Comparing MTL and MEL

In 2012, a higher proportion of MTL’s children were developmentally vulnerable in comparison to MEL’s children for almost all EDI indicators (DV1, DV2, and DV by domain), the only exception being in the PHYS domain where both settings show very similar results (see Table 4). For the four other domains, there were lower proportions of children developmentally vulnerable in MEL. These overall differences were replicated across sensitivity analyses with four other vulnerability thresholds (findings not presented). Differences between MTL and MEL were especially remarkable within the COGN domain (MTL 13.0% vs. MEL 5.8%) and the EMOT domain (MTL 11.5% vs. MEL 6.9%). The proportion of children vulnerable in at least one domain was 26.8% (95% CIs: 26.2, 27.3) in MTL, and 19.2% (95% CIs: 18.8, 19.5) in MEL. The same trend was found for the DV2 indicator (MTL = 13.2%, 95% CIs: 12.8, 13.6; MEL = 9.2%, 95% CIs:8.9–9.4). Moreover, median scores for each domain were systematically higher among MEL’s children. Again, the COGN and EMOT domains are those with the greatest differences between the two metropolitan cities. Interestingly, the COMM domain presents the biggest interquartile range between individual scores for both MTL and MEL.

### 3.3. Comparisons across Subgroups in MTL and MEL 

Overall, results showed some similar patterns of associations across subgroups of children in MTL and MEL: boys, immigrants (children born outside Canada or Australia), those who did not speak the language of the majority at home and those who lived in deprived areas were all at greater risk of being developmentally vulnerable (see Table 5 and Appendix A). Boys experienced almost twice the risk of girls for the DV1 indicator in MTL (RR = 1.77, 95% CIs: 1.70, 1.85) as well as in MEL (RR = 1.85, 95% CIs: 1.78, 1.92). In both metropolitan areas, the greatest RR between boys and girls were observed within the EMOT domain (MTL RR = 3.21, 95% CIs: 2.96, 3.47; MEL RR = 3.55, 95% CIs: 3.28, 3.84).

In MTL, children who spoke French at home (the most widely spoken language in Quebec) presented the lowest risk of being developmentally vulnerable in at least one domain (DV1 = 23.5%, 95% CIs: 22.9, 24.2), compared to English-speaking children (DV1 = 30.5%, 95% CIs: 29.2–32.0) and those speaking a language other than French or English (DV1 = 33.0%, 95% CIs: 31.8, 34.2) (see Table 5 and Appendix A). In MEL, the RR of being vulnerable for children who speak English at home (the most widely spoken language in Australia) were lower for all EDI indicators, compared to those who do not. For instance, the DV1 proportions were 16.3% (95% CIs: 16.0, 16.7) and 27.9% (95% CIs: 27.1, 28.7) respectively. Not surprisingly, the COMM domain is where the most striking differences were observed between language subgroups in both settings (MTL RD = 12.6%, 95% CIs: 11.6, 13.6; MEL RD = 13.1%, 95% CIs: 12.4, 13.8). When comparing each city’s vulnerability risks between children who spoke the language of the majority at home, and those who did not, disparities among subgroups are generally smaller in MTL than in MEL. Statistically significant differences between the two settings were observed in language subgroups for the DV1 indicator (MTL RR = 1.40, 95% CIs: 1.34, 1.46; MEL RR = 1.71, 95% CIs: 1.65, 1.77), for the DV2 indicator (MTL RR = 1.59, 95% CIs: 1.48, 1.70; MEL RR = 1.85, 95% CIs: 1.74, 1.95) and for the COGN domain (MTL RR 1.50, 95% CIs: 1.40, 1.61; MEL RR = 1.90, 95% CIs: 1.77, 2.04).

Immigrant children are at greater risk of being vulnerable for almost all indicators in both cities (see Table 5 and Appendix A). The only exceptions were in MEL for the PHYS and the SOC domains where the RRs and RDs between immigrant and non-immigrant children were not statistically different. Again, COMM is the domain where the biggest RR were observed in both metropolitan areas. When comparing immigrant children’s risk in each city, no significant differences were observed for DV1 (MTL RR = 1.31, 95% CIs: 1.24, 1.39; MEL RR = 1.44, 95% CIs: 1.36, 1.51). Among the PHYS domain, the risk of being vulnerable for immigrant children was higher in MTL (RR = 1.49, 95% CIs: 1.31, 1.70) compared to MEL (RR = 1.01, 95% CIs: 0.91, 1.13), but smaller in the COMM domain (MTL RR = 2.09, 95% CIs: 1.89, 2.32; MEL RR = 2.50, 95% CIs: 2.33, 2.69).

In both cities, there is a strong social gradient in the proportions of vulnerable children across quintiles of material deprivation, for all EDI indicators, with the highest proportions of vulnerability being systematically observed in the most deprived areas (see Table 5 and Appendix A). Socioeconomic disparities among area-level quintiles were of greater magnitude in MEL than MTL for all indicators. For instance, the RD between 1st and 5th quintiles of deprivation were statistically different across both cities for DV1 (MTL = 12.8%, 95% CIs: 11.2, 14.4; MEL = 20.7%, 95% CIs: 19.6, 21.9) and DV2 (MTL = 8.9%, 95% CIs: 7.7, 10.2; MEL = 12.8%, 95% CIs: 11.9, 13.7). The largest differences between the two metropolitan areas were observed for the COMM domain (MTL = RR 2.18, 95% CIs: 1.94, 2.46; MEL = RR 4.89, 95% CIs: 4.41, 5.49) as well as in the COG domain (MTL RR = 1.79, 95% CIs: 1.64, 1.97; MEL = RR 5.38, 95% CIs: 4.71, 6.15). 

## 4. Discussion

The aims of this research were to compare ECD outcomes among children in their first year of schooling between the metropolitan areas of Montreal and Melbourne, as measured by a series of EDI indicators, and to contrast disparities across demographic and socioeconomic groups between both cities. Overall, the proportion of children developmentally vulnerable in at least one domain (DV1) was 26.8% (95% CIs: 26.2, 27.3) in MTL and 19.2% (95% CIs: 18.8, 19.5) in MEL (see Table 4). More specifically, the EMOT domain and the COG domain were those where the largest differences were found between the two metropolitan areas, and only the PHYS domain showed no statistically significant differences. In terms of inequalities, it was found that the risk of being developmentally vulnerable as a function of home language and area-level deprivation subgroups were significantly smaller in MTL than in MEL.

To our knowledge, this is one of the first international comparisons of the EDI results at the metropolitan level. Because the cities of MTL and MEL are relatively similar in terms of demographic and economic characteristics, and the data were collected in the same year, this comparative work represents an interesting opportunity to investigate jurisdictional disparities in ECD outcomes. Furthermore, this study was based on two population-wide surveys using adaptations of the EDI, a population-based measure for which there is growing interest for international comparisons of ECD outcomes [28,53]. There was a very high participation rate in MEL and a rigorous weighting process in MTL [38], resulting in data sets that are representative of their respective populations. To minimize the risk of information bias, we used the same vulnerability cut-offs, the same exclusion criteria, and near-identical questionnaires. For the data collected in MTL, a robust translation process was used to produce a French version of the EDI questionnaire which was equivalent to the English version [38,51]. 

Such international comparisons inevitably include methodological limitations. The most important intergroup difference in terms of participants’ characteristics was in relation to MEL’s children being younger than MTL’s, with a three-month difference in participants’ mean age (see Table 1). This difference likely results in an underestimation of the observed gap between both cities because younger age is strongly associated with greater likelihood of developmental vulnerability [39]. Moreover, using 95% CIs comparisons to define statistical significance is known to produce overly conservative tests, which might also result in an underestimate of differences between MTL and MEL [63]. Because only limited data are collected on participants’ socioeconomic and demographic characteristics in the EDI questionnaire, some individual-level variables known to be strongly associated with ECD could not be captured in our analyses (e.g., family’s income, parents’ level of education) [1,64]. However, it seems unlikely that results would be entirely explained by socioeconomic or demographic intergroup differences solely.

Although datasets in MTL and MEL were harmonized to make each variable as comparable as possible, some methodological differences remained. For instance, the number of questions in the QSCDK and the AEDC was not the same. The education context in which both surveys took place was not identical either (e.g., school curriculum, classroom size, teachers’ academic degrees). It was impossible to evaluate if teachers’ rating were influenced by cultural differences across MTL and in MEL. In the last decade, various adapted versions of the original EDI have been used around the world and have showed generally very high convergent validity, including in Canada and in Australia [33,48,49,50,51]. We are confident that the methods we adopted allow for a valid comparison of ECD outcomes despite those limitations. 

The applied consequences of the significant differences in results between the two cities are of interest from a population health point of view. For example, if MTL improved ECD outcomes to reach MEL’s levels (i.e., improving DV1 from 26.2% to 19.2%), how much ‘vulnerability’ might have been avoided? Applied to the total population of children in their first-year of schooling in the metropolitan area of MTL (~40,000), the MEL lower DV1 proportions would result in about 3000 children not being developmentally vulnerable every year, with potential cost savings in the longer term [5]. This larger number of at-risk children in MTL undeniably translates into extra needs and costs for services (e.g., speech therapist) and probably into substantial long-term consequences as well (e.g., school failures & drop-outs) [5,51,65]. Besides, it is known that vulnerable children are often clustered in the most deprived local communities, therefore creating a supplementary burden of greater health and psychosocial needs in small areas [66,67]. One could also underline that MEL results are worrisome as well, given that almost one in five children is developmentally vulnerable in at least one domain. Developmental vulnerability represents tremendous consequences and costs for all societies, even though there is a largely accepted consensus that it is at least partially avoidable [5,30,68]. For instance, Kershaw et al. (2010) argues that only 5% of children are born with a development-impairing problem and that reducing proportions of vulnerable children to 10% is a conservative target that all jurisdictions should aim to meet [5]. 

It appears important to address the question of why two jurisdictions whose populations share similar demographic and socioeconomic characteristics result in such different proportions of developmentally vulnerable children (i.e., respectively 7.6 and 4.0 percentage point differences between MTL and MEL for DV1 and DV2, see Table 4). Although it would be impossible at this point to identify specific sociopolitical or community-level factors for explaining the differences we observed, it seems unlikely they are exclusively genetic or cultural per se. Therefore, we believe it relevant to put these findings into perspective with some noteworthy child programs and family policies implemented in each jurisdiction. It is reasonable to hypothesize that overall EDI results are more likely explained by the positive effects of universal programs rather than the effects of programs targeting only low-income families, as the majority of children vulnerable in at least one domain are not from low socioeconomic status but rather from the middle class [36]. To explain the overall better results in MEL (DV1 = 19.2%, 95% CIs: 18.8, 19.5; DV2 = 9.2%, 95% CIs: 8.9, 9.4) compared to MTL (DV1 = 26.8%, 95% CIs: 26.2, 27.3; DV2 = 13.2%, 95% CIs: 12.8, 13.6), one could hypothesise that the Victoria ‘10 key ages and stages’ visits with a Maternal and Child Health nurse may have been beneficial for improving ECD outcomes. This is a unique universal program covering a large range of child health and development issues (e.g., growth monitoring, developmental assessment, counseling on healthy behaviors and parenting practices) from birth to 3.5 years-old [36,69]. In Montreal (and in Quebec), there is no such structured preventive programs for young children. Access to periodic health and development assessments and to counseling relies solely on primary care physicians, for which access can sometimes be difficult [70]. The national universal preschool access reform in Australia might also have had a role in explaining MEL’s results, as attendance has been associated with reduced risk of being developmentally vulnerable for children living in both advantaged and disadvantaged communities of Australia [71]. It is well known that preschool programs can benefit young children’s development by providing a rich learning environment which supports development of language, cognition, and socioemotional skills [71,72,73,74]. Unfortunately, it has been shown that in Australia and elsewhere, disadvantaged children are less likely to attend any form of early education services than other children, and when they do, they are more likely to attend lower quality services [75,76,77]. This could contribute to the inequalities that were found in this study.

This study also shows similar patterns of associations among subgroups of children in MTL and MEL. Boys, immigrants, children who do not speaking the language of the majority at home and those living in the most deprived areas were all at greater risk of being developmentally vulnerable (see Table 5). These results are not surprising, as the vast majority of studies using the EDI have shown these risk trends [44,45,47,78,79,80]. However, language and area-level deprivation subgroup results showing fewer disparities in MTL compared to MEL concerning DV1 and DV2 were not anticipated. For instance, RR between 1st and 5th quintiles of deprivation are remarkably different in both cities for DV2 (MTL RR = 1.95, 95% CIs: 1.78, 2.14; MEL RR = 3.82, 95% CIs: 3.48,4.20). Interestingly, compared to other Canadian provinces, Quebec is often recognized for its generous social safety net for the most disadvantaged families and innovative investments in child and family policies [81,82,83]. In particular, the implementation of the well subsidized universal early education centres program known as ‘Centres de la petite enfance’ (CPE) in 1997 has led to substantial increases in regulated childcare use, and to greater use of these services by low-income families than elsewhere in Canada [81,82]. On average, quality criteria are more likely to be met in CPE than in private childcare services according to different surveys in the province [77,84,85,86]. Furthermore, Laurin et al. (2016) demonstrated that exclusive attendance to a CPE was beneficial for children from low-income families in significantly reducing their risk of being developmentally vulnerable, compared to other early educational pathways [87]. However, from 2009 to 2016, the proportion of total daycare spaces in CPEs dropped significantly in Quebec because of changes in governmental policies, i.e., new tax credits and increased permit deliverance for private daycare services [88]. In 2012, only one in three children having attended early educational services had exclusively attended a CPE mainly because of lack of available spaces [89]. In MTL, access to CPE remains a problem for a majority of families, but some programs targeting disadvantaged children aim to integrate them in priority at low or no cost [87,88,90]. In Australia, ensuring the most deprived children have access to high-quality early education services seems to be more of a challenge that may be contributing to the greater inequities we observed in MEL compared to MTL [71,75,76]. 

The fact remains that both MTL and MEL face unfair and avoidable subgroup disparities in all EDI indicators, as is the case in many settings throughout the world [91,92]. To reduce the inequity gap in ECD and better reach out to at-risk families, all dimensions of access to and quality of services must be addressed [93]. Particularly in the field of primary health care and early education programs, new innovative strategies targeting financial, geographical, and sociocultural access barriers are needed to achieve genuine proportionate universalism to a range of high-quality early years services [2,30,94,95].

## 5. Conclusions

This international comparison of ECD outcomes at the metropolitan level points to intriguing similarities and differences between MTL and MEL metropolitan areas. MEL’s children globally experience better ECD outcomes than MTL’s children. However, there were smaller disparities in MTL compared to MEL among the language and area-level deprivation subgroups. This study provides the first opportunity to learn more from each metropolitan jurisdiction to better inform local, regional, and national policies for young children. Further research is warranted to identify the environmental factors, policies and programs that account for these observed differences. A systematic review of universal and targeted services and policies for young children in each setting may provide a better understanding of the results, particularly in the field of early education and primary health care. Future international comparisons using the EDI may lay the ground for considering how jurisdictions could test action-oriented policy interventions to improve early childhood programs and services.

## Figures and Tables

**Table 1 ijerph-16-02915-t001:** Characteristics of populations in the metropolitan Montreal and Melbourne in 2011 (Information extracted from [21,22] respectively).

Characteristics of Populations	Greater Montreal	Greater Melbourne
Total population	3.93 million	4.17 million
Area	4604 km^2^	9990 km^2^
Population density	890.2 persons/km^2^	417.3 persons/km^2^
Home Language	16.5% English 65.2% French 18.3% Other	70.9% English 29.1% Other
Immigrant population	25.1%	36.7%
Indigenous population	0.9%	0.5%
Unemployment rate	7.5%	5.5%
Smaller geographical areas	Greater MTL: 82 municipalities City of MTL: 19 districts	Greater MEL: 31 LGA City of MEL: 15 suburbs

LGA: local government area. MTL = Montreal. MEL = Melbourne.

**Table 2 ijerph-16-02915-t002:** Data harmonization to account for survey methodological differences.

Survey Differences	Montreal (MTL)	Melbourne (MEL)	Data Harmonization
Questionnaire	- QSCDK 2012 - 104 questions - Identical to the original EDI, translated to French - Translation-backtranslation for each item	- AEDC 2012 - 95 questions - 9 questions removed from the original EDI - 1 minor wording change (e.g., bathroom vs. washroom)	- Scores by domain based on the original questionnaire of each setting - High internal consistency after the changes in the AEDC questionnaire
Vulnerability cut-offs	- QSCDK 2012 - Not age-based	- AEDC 2009 - Age-based for 4-, 5-, 6-year-old children	- All QSCDK indicators were recalculated based on the AEDC age-based 2009 cut-offs - Sensitivity analyses with different Canadian and Australian cut-offs
Participation rate	- 75.9% - Weighting process to adjust for population characteristics	- 97.4% - No weighting	- Sensitivity analyses with and without the weighting variable in MTL
Exclusion criteria	- Students with handicaps, social maladjustments or learning difficulties (SHSMLD), as defined by the Quebec Ministry of Education [60]	- Children with special needs (SN), as defined in the AEDC data dictionary [61]	- Exclusion criteria based on each setting’s categories were kept after in-depth examination of diagnoses included in SHSMLD and SN
Language	- Home language in 3 categories (French, English, other)	- Home language in 2 categories (English, other)	- 3 categories in MTL - 2 categories in MEL - Risk comparison between the best and worst outcome language categories in MTL
Area-level deprivation	- Quintiles from the MSDI - Based on Dissemination Area, the smallest statistical area level in Canada - Quebec is the reference population	- Quintiles from the Index of Relative Socio-economic Disadvantage (IRSD) - Based on the smallest statistical area in Australia - Australia is the reference population	- Quintiles of Area-level disadvantage based on each setting’s own index - For Australia, the quintiles were recalculated according to the Victoria IRSD scores

QSCDK: Quebec Survey of Child Development in Kindergarten. AEDC: Australian Early Development Census. EDI: Early Development Instrument. SHSMLD: Students with handicaps, social maladjustments or learning difficulties. SN: Special needs children. MSDI: Material and Social Deprivation Index. IRSD: Index of Relative Socio-Economic Deprivation.

**Table 3 ijerph-16-02915-t003:** Characteristics of the surveyed children in the metropolitan Montreal and Melbourne areas.

Characteristics	Montreal (MTL)	Melbourne (MEL)
Participating children (*N*)	29,391	51,009
Sex (%)		
Boys	49.8%	50.3%
Girls	50.2%	49.7%
Age in months (mean)	72.1	69.1
Age groups (%) <69 months 69 months to <72 months 72 months to <75 months ≥75 months	23.9% 25.9% 24.1% 26.0%	46.1% 24.8% 18.9% 10.2%
Country of birth (%) In home country Outside home country	90.1% 9.9%	91.3% 8.7%
Home language (%)	French 61.4% English 14.9% Other than French/English 23.7%	English 75.3% --- Other than English 24.7%
Area-level deprivation * (%) ** Q1 Q2 Q3 Q4 Q5	27.0% 21.7% 16.3% 16.1% 19.0%	23.3% 22.3% 20.0% 17.7% 16.9%

* Quintile of deprivation as compared to Quebec’s population for MTL and Victoria’s population for MEL. ** Q1: least deprived quintile. Q5: most deprived quintile.

**Table 4 ijerph-16-02915-t004:** Proportions of developmentally vulnerable children and median scores in Montreal and Melbourne.

Proportions and Median Scores	DV1	DV2	PHYS	SOC	EMOT	COGN	COMM
**Montreal (MTL)**							
% vulnerable	**26.8%**	**13.2%**	7.3%	**9.9%**	**11.5%**	**13.0%**	**8.9%**
(CI)	(26.2–27.3)	(12.8–13.6)	(7.0–7.6)	(9.6–10.3)	(11.2–11.9)	(12.6–13.4)	(8.5–9.2)
Median scores			9.6	9.0	8.1	8.8	9.4
(IQR)	---	---	(1.2)	(2.1)	(2.0)	(1.5)	(2.5)
**Melbourne (MEL)**							
% vulnerable	**19.2%**	**9.2%**	7.4%	**7.9%**	**6.9%**	**5.8%**	**8.0%**
(CI)	(18.8–19.5%)	(8.9–9.4%)	(7.2–7.6%)	(7.7–8.2)	(6.7–7.2)	(5.6–6.0)	(7.8–8.3)
Median scores (IQR)	---	---	10 (1.3)	9.4 (1.9)	8.8 (2.0)	9.6 (1.2)	10 (2.5)

DV1: Developmentally vulnerable in at least 1 domain. DV2: Developmentally vulnerable in at least 2 domains. PHYS: Physical Health and Well-Being. SOC: Social Competence. EMOT: Emotional Maturity. COGN: Language and Cognitive Development. COMM: Communication Skills and General Knowledge. CI: 95% Confidence interval. IQR: Interquartile range. Bold: statistically significant difference between MTL and MEL, by comparing CI intervals.

**Table 5 ijerph-16-02915-t005:** Proportions of developmentally vulnerable children for overall indicators, relative risks and risk differences across subgroups in the metropolitan Montreal and Melbourne.

Subgroups of Children	DV1		DV2	
	MTL	MEL	MTL	MEL
Sex				
Girls	19.3% (18.7–20.0)	13.4% (13.0–13.9)	18.3% (17.6–18.9)	12.6% (12.2–13.0)
Boys	34.2% (33.4–35.0)	24.9% (24.3–25.4)	8.2% (7.7–8.6)	5.7% (5.4–6.0)
RR	1.77 (1.70–1.85)	1.85 (1.78–1.92)	2.24 (2.10–2.40)	2.20 (2.07–2.33)
RD	**14.9% (13.8–15.9)**	**11.4% (10.8–12.1)**	**10.1% (9.3–10.9)**	**6.9% (6.4–7.4)**
Home language				
French	23.5% (22.9–24.2)	---	11.1% (10.6–11.6)	---
English	30.5% (29.2–32.0)	16.3% (16.0–16.7)	14.8% (13.8–15.9)	7.6% (7.3–7.9)
Other	33.0% (31.8–34.2)	27.9% (27.1–28.7)	17.6% (16.6–18.6)	14.0% (13.4–14.6)
* RR	**1.40 (1.34–1.46)**	**1.71 (1.65–1.77)**	**1.59 (1.48–1.70)**	**1.85 (1.74–1.95)**
RD	**9.4% (8.1–10.7)**	**11.6% (10.7–12.5)**	6.5% (5.4–7.6)	6.4% (5.8–7.1)
Country of birth				
Home country	25.4% (24.9–26.0)	18.5% (18.1–18.8)	18.1% (16.6–19.6)	12.6% (11.6–13.6)
Outside home country	33.4% (31.5–35.3)	26.5% (25.3–27.8)	12.4% (12.0–12.9)	8.9% (8.6–9.1)
RR	1.31 (1.24–1.39)	1.44 (1.36–1.51)	1.45 (1.33–1.59)	1.42 (1.31–1.55)
RD	8.0% (6.0–9.9)	8.1% (6.7–9.4)	5.6% (4.0–7.2)	3.7% (2.7–4.8)
Area-level deprivation				
** Q1	21.2% (20.3–22.2)	11.9% (11.4–12.5)	9.4%. (8.7–10.1)	4.5% (4.2–4.9)
Q2	24.2% (23.1–25.3)	14.8% (14.2–15.5)	11.6% (10.8–12.5)	6.8% (6.3–7.3)
Q3	27.9% (26.6–29.3)	18.0% (17.3–18.8)	13.8% (12.8–14.9)	8.6% (8.0–9.1)
Q4	29.6% (28.2–31.0)	22.6% (21.8–23.5)	15.0% (14.0–16.1)	11.2% (10.6–11.9)
Q5	34.0% (32.8–35.3)	32.7% (31.7–33.7)	18.3% (17.3–19.4)	17.3% (16.5–18.1)
RR	**1.61 (1.51–1.70)**	**2.74 (2.59–2.90)**	**1.95 (1.78–2.14)**	**3.82 (3.48–4.20)**
RD	**12.8% (11.2–14.4)**	**20.7% (19.6–21.9)**	**8.9% (7.7–10.2)**	**12.8% (11.9–13.7)**

(95% CIs). RR: Relative risk. RD: Risk Difference. Bold: statistically significant differences in RR or RD between MTL and MEL. * For MTL: French vs. Other. For MEL: English vs. Other. ** Q1: least deprived quintile. Q5: most deprived quintile. RR and RD were calculated using Q1 vs. Q5. Bold: statistically significant differences between MTL and MEL, by comparing CI intervals.

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
