# Peer review of "Does Place Matter? An International Comparison of Early Childhood Development Outcomes between the Metropolitan Areas of Melbourne, Australia and Montreal, Canada"

_ijerph, 2019, doi:10.3390/ijerph16162915_

Round 1
Reviewer 1 Report
The paper presents a comparison on early child development, between to different contexts, one in Canada and another in Australia. Similar and different characteristics are shared.
Perhaps “keywords” could be reviewed and could include the contexts of research, as it was so relevant in the paper writing, e.g. in the title of the article.
Abstract/ Methods: identify and describe the data analysis and software programs used to support the research.
Table 1 has no source, please identify it.
It referred the importance of international comparisons of ECD outcomes. But it is not clear, what were the criteria for the selection: Melbourne and Montreal? Is it just the similar socioeconomic and demographic matter?
It is expected to see deeper discussion and a more explored “conclusion”. Here, a possibility, to better inform the reader, is to integrate some details supported by the results as the evidence that Melbourne's children globally experience better ECD outcomes than Montreal’s children, but that inequity gaps are greater in Melbourne, and for instance some implications for practice.
Looking at “References”, it is possible to find some links from Wikipedia, however it is not cited in the main text. Wikipedia is a collaborative website with free entrances with any supervision. There is no author or published date. Papers should generally rely on peer-reviewed and other scholarly work vetted by experts in the field. I suggest removing it all and instead please look for reliable source.
Author Response
Thank you for you kind suggestions, please find my responses in the attachment.

Reviewer 2 Report
As the authors mentioned in the paper, international comparison is a valuable study. However, the authors need a clearer explanation and detailed procedure on their method.
One important point the authors need to explain is about the difference of questionnaires between two countries. MEL removed 8 questions and added 1 question. The authors need to explain the reason why this modification has been applied. Also, the authors need to prove that all domains (or factors) are not affected by this modification to ensure the validity of the different questionnaires between two countries. They need to show some analysis results such as exploratory factor analysis to ensure MEL questionnaire are the same as the original version and MTL version.
The authors need to explain the results of Table 5 more clearly. They need to list the 95 confidence intervals showing which values are compared to come up with statistically significant results.
The authors need to eliminate some bias. The first bias that I found is questionnaire differences. The validity for two language questionnaires should be reported. The second bias is one caused by cultural differences. How can you explain that the difference is totally from the children’s experience and not from two different culture? Cultural difference has to be controlled before comparing two cities inequity gap.
Some variable are not considered in the comparison of two cities: education setting (such as school types, school curriculums, classroom size, teachers’ background and quality, etc. ), SES of families, and demographic information ( family types, caregivers ‘ education or career background, etc. ). Since these variables could be one of causes that might change the result of children’s achievement and/or developmental skills, they should be carefully considered or controlled.
Author Response
Thank you for your kind suggestions, please find my responses in the attachment.
Round 2
Reviewer 2 Report
I strongly believe that this study will bring a great benefit to the field of Early Childhood Education because it is, as the authors stated, the first research of international comparison of Early Childhood Development Outcome. However, I still have some questions remained on their research method.
After reviewing this paper, I did some research on my own about international comparison studies in Education field. One example that I found was OECD’s Program for International Student Assessment (PISA). When researchers make an international comparison of older students, such as in PISA, they develop assessment tools together from the beginning of their research. They develop assessment questions together, test their translation process if the test needs to be translated, adjust/control any cultural or social differences if they found students might have confusion, and perform the assessment with same age groups or grade levels. This process helps researchers to reduce the validity, reliability, and accuracy issues of a comparison study.
What I found questionable about this reviewed research was that the authors compared two different assessments that were used separately. Two different nations did not work together to plan for a test or to adjust the assessment tools. I have never seen a study that the researchers would just compare and contrast the outcomes of two differently performed tests. Even though the authors insist that two tests are comparable, I still could not be 100% sure if these two assessment tools could be compared. I am not an expert on educational statistics, so I asked very basic and brief questions that I had when I reviewed this article. I only commented on some methodical points that I found questionable when I reviewed this paper for the first time. As I have read the authors’ responses, I am still not confident in saying OK to the research method part. I recommend you to send this article one more time to an educational statistics expert for a final review, if possible.
Author Response
Response to reviewer 2 – round 2
Manuscript IJERPH-509604
Reviewer’s comment:
What I found questionable about this reviewed research was that the authors compared two different assessments that were used separately. Two different nations did not work together to plan for a test or to adjust the assessment tools. I have never seen a study that the researchers would just compare and contrast the outcomes of two differently performed tests. Even though the authors insist that two tests are comparable, I still could not be 100% sure if these two assessment tools could be compared. I am not an expert on educational statistics, so I asked very basic and brief questions that I had when I reviewed this article. I only commented on some methodical points that I found questionable when I reviewed this paper for the first time. As I have read the authors’ responses, I am still not confident in saying OK to the research method part. I recommend you to send this article one more time to an educational statistics expert for a final review, if possible.
Authors’ response:
Ø see minor modifications in lines 171-174, and lines 328-329 of the manuscript
We think it is very relevant to ask such questions about methodological differences in the reviewing process of international comparison studies. It is true that the measurement of ECD outcomes may be influenced by culture and language, making international comparison complex. To our knowledge, when a validated instrument is well implemented in many countries, as it is the case for the EDI, it is legitimate to use it retrospectively to compare outcomes in different settings, even though the two nations did not work together to plan and adjust the questionnaire. In the case of Quebec and Australia, prior to compare the EDI across countries, validity studies have been conducted for each adapted version. As the EDI provides a well validated population-based measure at relatively low cost, there is increasing support from different organisations such as the World Bank for its use at the national level, but also as a recommended test for international comparisons of ECD [1-4].
1. Young M, Richardson L. Early Child Development from Measurement to Action: The World Bank; 2007. 322 p.
2. Fernald LCH, Prado EP, Kariger P, Raikes A. A Toolkit for Measuring Early Childhood Development in Low- and Middle-Income Countries Washington DC: The World Bank; 2017.
3. Janus M, Harrison LJ, Goldfeld S, Guhn M, Brinkman S. International research utilizing the Early Development Instrument (EDI) as a measure of early child development: Introduction to the Special Issue. Early Childhood Research Quarterly. 2016;35:1-5.
4. Brinkman SA, Gialamas A, Rahman A, Mittinty MN, Gregory TA, Silburn S, et al. Jurisdictional, socioeconomic and gender inequalities in child health and development: analysis of a national census of 5-year-olds in Australia. BMJ Open. 2012;2(5).